# Export–Output Growth Nexus Using Threshold VAR and VEC Models: Empirical Evidence from Thailand

**Arisara Romyen** [1,2,*] **, Jianxu Liu** [3,4] **and Songsak Sriboonchitta** [1,4]

1   Faculty of Economics, Chiang Mai University, Chiang Mai 50200, Thailand; songsakecon@gmail.com
2   Faculty of Economics, Prince of Songkla University, Songkhla 90110, Thailand
3   Faculty of Economics, Shandong University of Finance and Economics, Jinan 250000, China; liujianxu1984@163.com
4   Puey Ungphakorn Center of Excellence in Econometrics, Faculty of Economics, Chiang Mai 50200, Thailand
*   Correspondence: arisara.r@psu.ac.th

**Abstract:** This paper explores the relationship between export, import, and output for Thailand over the period from 1990 to 2017. The threshold vector autoregressive (VAR) and threshold vector error correction (VEC) models were applied. The empirical evidence confirms that the export-led growth hypothesis is valid, implying feedback within the export–output growth nexus. During business cycles, the export–output characteristics in economic cycles can be classified by the two-threshold VAR and VEC models. These relevant variables converge from the long-run equilibrium. As for the thresholds which are correlated, gross domestic product (GDP) vs. export and GDP vs. import exist as a long-run equilibrium relationship, while there does not seem to be a relationship of export vs. import. Furthermore, a five-year forecast was created (the period of 2018–2022). The export–output growth scenarios appear to swing upward continuously throughout the short-term trend. Therefore, policy-makers should highlight countercyclical macroeconomic policies at lower, medium, and upper regimes to strengthen the state of recovery and encourage the state of short recession.

**Keywords:** non-linear time-series models; export–import pass-through; threshold autoregression; structure break; predictive threshold VAR model

**JEL Classification:** F14; O04

## 1. Introduction

The role of outward-orientation policies in economic growth for developing countries was an important issue for debate in the past few decades. Their comprehensive consequences remain a great controversy. Thailand recently enhanced and promoted intensive advancement in the export–output growth nexus. As a result of policies favoring outward-oriented growth, Thailand encounters many challenges, especially during global economic crises such as the 1997 Asian financial crisis or the 2007 sub-prime mortgage recession in the United States (US) that spread at a rapid pace to the economy. Therefore, Thailand's export–output nexus is connected to the consequences of external and internal structure transformation. Figure 1 shows the performances of the export, import, and gross domestic product (GDP) of Thailand from 1990 to 2017. The figure also demonstrates that all series fluctuate and may not be stationary due to regime shifts and structural changes. Particularly in the cases of the 1997 Asian financial crisis and the 2007 sub-prime mortgage crisis in the US, it was found that these shocks remarkably affected Thailand's export–output nexus. Circumstances such as these may cause high volatility.

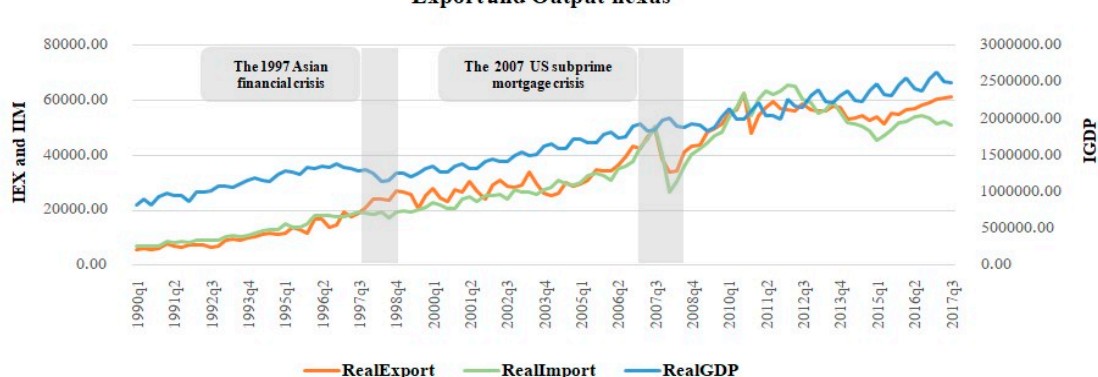

**Figure 1.** Export and output growth nexus in Thailand.

An export-led growth (ELG) scheme was initially scheduled as a strategy encouraging the export to contribute economic growth and international integration (Balassa 1971). The hypothesis of ELG is verified by the positive attribution from export toward growth through its multiplier effects (Krueger 1978; Ram 1985; Salvatore and Hatcher 1991; Sengupta 1993; Ulrich 2014). Trade openness is considered as an important engine of economic expansion since it can promote a potential allocation of resources and can offer several dynamic benefits. On the other hand, a growth-led export (GLE) scheme means that economic growth for a country can promote its export volume and can bring more products to offer domestic demand. The hypothesis of GLE analyzes the relationship between economic growth and exports. According to the GLE hypothesis, the causality direction from economic expansion into export can occur (Arnade and Vasavada 1995), or bi-directional causality can also exist (Kugler and Dridi 1993; Xu 1996).

Many studies used the conventional vector autoregressive (VAR) model to examine the casual nexus between export and output, which may undergo obstacles due to business cycles. Given the presence of structural breaks, linear modeling techniques may provide an imprecise analysis since the assumption of symmetry in the dynamic equilibrium residuals is impractical. Hence, a cointegration test based upon a linear VAR (one regime) is ineffective during the transition from boom to bust, or vice versa (Lee and Huang 2002; Haddad 2010). To address this issue, we apply the threshold regression model, which was developed by Hansen (2000) to deal with the nonlinear effect on the export–output nexus. The threshold model approach can capture the process instantaneously through the business cycle, which has distinct patterns such as recessions, recoveries, and ordinary growth. Each phase performs its unique dynamics; this structural change directly affects the international business participants in the global economies. Moreover, policy-makers cannot entirely anticipate the direction of such market-driven search processes. Therefore, we form the regime of the economy as a latent business cycle variable along with the threshold autoregression structure using observed economic variables. Ultimately, we can create more precise predictions, especially at turning points, and analyze the effect of dynamics across business cycle phases.

The aims of this paper were as follows: (1) to estimate the relationship between the export, import, and output for Thailand by taking into account the threshold levels, with an attempt to compare the nonlinear models within the thresholds corresponding to the transition from economic cycles; and (2) to predict the export–output nexus over a five-year period. The contribution of this paper is to determine the presence of more than two regimes. The article is organized as follows: Section 2 introduces the necessary background of the nonlinear model. Section 3 presents the econometric methodology. Empirical findings are presented in Section 4. Finally, Section 5 provides the summary and policy recommendations.

## 2. Background

*Relationship between Export and Economic Growth*

A large number of empirical studies were previously carried out on the causal effect of outward-oriented trade policies on economic growth. Edwards (1993) classified these studies into three critical viewpoints. Firstly, initial studies relied on bivariate correlation analysis, ignoring the key factors on growth which were offered by the neoclassical growth theory, such that a spurious relationship could be found. Secondly, previous work paid no attention to the reverse flow of causality in terms of export and growth (and vice versa) due to the limitation of dynamic time-series databases. Thirdly, statistical adequacy and valid statistical models, such as the assumption of a linear regression between export and growth, seemed to be an inaccurate inference. More recent ELG studies were devoted to the neoclassical production approach by adopting sophisticated econometric models (Yamada 1998; Awokuse and Christopoulosb 2009; Liu and Zhang 2015).

Later on, time-series methods such as cointegration and error correlation modeling techniques were applied to investigate the appearance of long-run cointegrating correlations in terms of ELG hypothesis (Richards 2001; Reppas and Christopoulos 2005). Thereby, the nonlinearity model in the export and output nexus was moderately disregarded in the literature. The early empirical evidence followed either the ELG or GLE hypothesis, regularly neglecting the time-varying dynamic correlation between the series properties. According to structure break persistence, a causal relationship between the relevant variables is likely misleading, since this causality might hold only a part of the whole. Therefore, nonlinear dynamics for macroeconomic data (e.g., GDP growth, export and import, exchange rate) become necessary tools in the case of business cycles.

## 3. Methodology

Under global economies, it is essential to detect structural change and to facilitate several stakeholders for business arrangements. To fill this role, the threshold model is specified among several nonlinear models since it is a straightforward generalization toward linear models. For instance, the threshold vector autoregressive (TVAR) model identifies distinct autoregressive patterns through each stage, as well as a threshold variable, which considers each regime as active. Hence, these models are more appropriate for predicting nonlinear models, especially hidden Markov models (Tsay 1998; Hubrich and Teräsvirta 2013). The threshold vector autoregression model and the threshold vector error correction models are described in Sections 3.1 and 3.2, respectively. The generalized impulse response function is presented in Section 3.3, and the econometric methodology is detailed in Section 3.4.

*3.1. Threshold Vector Autoregression Model (TVAR)*

The two-threshold VAR model can be written as follows:

$$y_t = \begin{cases} \alpha_1 + A_1(L)y_t + \varepsilon_{1t}, \ if \ \gamma_1 < q_t \\ \alpha_2 + A_2(L)y_t + \varepsilon_{2t}, \ if \ \gamma_1 < q_t < \gamma_2, \\ \alpha_3 + A_3(L)y_t + \varepsilon_{3t}, \ if \ q_t < \gamma_2 \end{cases} \tag{1}$$

where $y_t$ is the vector of endogenous variables, and $q_t$ is the vector of threshold variables. $\gamma_1$ and $\gamma_2$ stand for the threshold variables. $\alpha_1$, $i = 1, 2$ and 3, is the vector of constants, whereas the lag-polynomial $A_i(L) = A_{i1}(L) + A_{i2}(L^2) + \ldots + A_{ip}(L^{2p})$, with $4 \times 4p$ matrix, where $A_{ij}$, $J = 1, 2, \ldots, p$, and $L$ is the lag operator. A compact form of the TVAR ($p$) model can be expressed as follows:

$$y_t = \begin{cases} \left(\alpha_1 + A_{11}(L)y_{t-1} + \ldots + A_{1p}(L)y_{t-p} + \varepsilon_{1t}\right)I\left(q_t \leq \gamma_1\right) \\ \left(\alpha_2 + A_{21}(L)y_{t-1} + \ldots + A_{2p}(L)y_{t-p} + \varepsilon_{2t}\right)I\left(\gamma_1 < q_t \leq \gamma_2\right) \ , \\ \left(\alpha_3 + A_{31}(L)y_{t-1} + \ldots + A_{3p}(L)y_{t-p} + \varepsilon_{3t}\right)I\left(q_t > \gamma_2\right) \end{cases} \tag{2}$$

where $I(.)$ obtains the value 1 in case its argument is satisfied and 0 otherwise. Given $\theta = (\alpha_1, \alpha_2, \alpha_3, A_1, A_2, A_3, \gamma_1, \gamma_2)$ as the vector of relevant parameters, the ordinary least square is employed to minimize the following function:

$$\theta = \arg min \left( \sum_{t=1}^{T} \left( \begin{array}{c} y_t - \left(\alpha_1 + A_{11}(L)y_{t-1} + \ldots + A_{1p}(L)y_{t-p} + \varepsilon_{1t}\right)I(q_t \leq \gamma_1) \\ -\left(\alpha_2 + A_{21}(L)y_{t-1} + \ldots + \quad A_{2p}(L)y_{t-p} + \varepsilon_{2t}\right)I(\gamma_1 < q_t \leq \gamma_2) \\ -\left(\alpha_3 + A_{31}(L)y_{t-1} + \ldots + A_{3p}(L)y_{t-p} + \varepsilon_{3t}\right)I(q_t > \gamma_2) \end{array} \right) \right) \tag{3}$$

*3.2. Threshold Vector Error Correction Model (TVECM)*

Threshold vector error correction models, which were initially proposed by Balke and Fomby (1997), are applied to investigate the adjustment of relevant variables to the long-term equilibrium. For two variables, $y'_t = (y_{1,t}, y_{2,t})$, they shift differently between regimes as follows:

$$\Delta y_t = u_t + F(e_{t-1})I + \sum_{i=1}^{k} \gamma_k \Delta y_{t-k} + u_t; \ t = 1, \ldots, n, \tag{4}$$

where $F(e_{t-1}) = \delta \alpha_1 e_{t-1} + (1-\delta)\alpha_2 e_{t-1}$ and $\delta = \begin{cases} 1 \text{ if } e_{t-1} \leq \lambda_1 \\ 0 \text{ if } e_{t-1} > \lambda_2 \end{cases}$. The $F(\cdots)$ defines an indicator function of the error correction ($e_{t-1}$), which is assigned as zero mean with stationary covariance. Vectors $\alpha'_i = (\alpha_{1,i}, \alpha_{2,i})$ and $\gamma'_i = (\gamma_{1,i}, \gamma_{2,i})$ indicate adjustment and short-term equilibrium parameters. The TVECM error ($u_t$) is supposed to be an independent and identically distribution or *i.i.d.* Gaussian sequence associated with a finite covariance matrix, $\Sigma = E\{u_t u'_t\}$.

Hansen and Seo (2002) (hereinafter referred as HS) developed a quasi-maximum likelihood estimations (MLE) approach based on a grid search through the cointegrating vector and the threshold parameter accompanied by a fixed-repressor bootstrap to explore threshold effects in a two regime model, as shown in Equation (4) with $\lambda_1 = \lambda_2 = \lambda$ and $\beta_0 = \beta_2 = 0$. The HS speculates the following likelihood function:

$$L\left(\alpha'_i, \gamma'_i, \beta_1, \Sigma, \lambda\right) = -\frac{n}{2}\log|\Sigma| - \frac{1}{2}\sum_{t=1}^{n} u_t\left(\alpha'_i, \gamma'_i, \beta_1, \Sigma, \lambda\right)' \Sigma_{t=1}^{n-1} u_t\left(\alpha'_i, \gamma'_i, \beta_1, \Sigma, \lambda\right). \tag{5}$$

The maximization of the likelihood function exists in case the parameters $\left(\alpha'_i, \gamma'_i, \Sigma\right)$ are related by assuming the parameters $(\beta_1, \lambda)$ are fixed, thereby obtaining a constrained maximum-likelihood estimator of $\Delta y_t$ on $e_{t-1}$, as well as $\Delta y_{t-1}, \ldots, \Delta y_{t-k}$, for each fixed $\beta_1$ and $\lambda$ presented. Then, the concentrated likelihood can be yielded as follows:

$$L(\beta_1, \lambda) = L\left(\beta_1, \lambda, \hat{a}'_i(\beta_1, \lambda), \hat{\gamma}'_i(\beta_1, \lambda), \hat{\Sigma}(\beta_1, \lambda)\right) = -\frac{n}{2}\log\left|\hat{\Sigma}(\beta_1, \lambda)\right| - n. \tag{6}$$

The maximum-likelihood estimators $(\hat{\beta}_1, \hat{\lambda})$ minimize $\log\left|\hat{\Sigma}(\beta_1, \lambda)\right|$ with respect to the probability $P$ that observations lie outside of the following threshold:

$$\pi_0 \leq P(e_{t-1} \leq \lambda) \leq 1 - \pi_0, \tag{7}$$

where $\pi_0$ is a trimming parameter. The algorithm provides sub-regions of the parameter space for $\beta_1$. The null hypothesis of linear cointegration is tested versus the alternative of threshold cointegration by means of the Supremum Lagrange multiplier (sup-LM) test following Andrews and Ploberger (1994), as shown below.

$$SupLM = \sup_{\lambda_1 \leq \lambda \leq \lambda_2} LM\left(\widetilde{\beta}_1, \lambda\right). \tag{8}$$

### 3.3. Generalized Impulse Response Function (GIRF)

The threshold impulse response function describes how changes of the system due to a shock rely on the past history within each regime specified by the estimated threshold. These impulse responses correspond to a regime-dependent characteristic, in which they are conditional in the regime prevailing throughout the horizon of the response. For a non-linear form, the reactions of endogenous variables due to the shock rely on the past history under the interesting period. The shock at time *t* may provoke a shifting of regime until time *t* + *d*, where *d* denotes the estimated lag of the threshold. To analyze impulse responses, the characteristic of a nonlinear time series is actually sensitive to the value of the parameters since even tiny changes in the parameters can vary the limiting dynamic behavior of the difference equation toward a unique figure of disturbance. As a sequence, it leads to invalid estimations. Determinations of persistence and asymmetry in response are carried out with several types of time series. The generalized impulse response function (GIRF) of the nonlinear time series is further extended through the standard impulse response function (IRF). The GIRF can handle an invariant linear structure on the time series. The dynamic properties of nonlinear stochastic aspect are incorporated with a probability measure in the sample path section applying the zero-innovation condition. Since nonstationarity is anticipated by the specific sample path restriction, then this measure is more manifest than the IRF. Koop et al. (1996) proposed the GIRF under alternative regimes, which were constructed in order to characterize the influences of historical shocks. The GIRF takes the form of

$$\text{GIRF} = E\Big[Y_{t+d}\big|\varepsilon_t, \varepsilon_{t+1} = 0, \dots, \varepsilon_{t+d} = 0, \Omega_{t-1}\Big] - E\Big[Y_{t+d}\big|\varepsilon_t = 0, \varepsilon_{t+1} = 0, \varepsilon_{t+d} = 0, \Omega_{t-1}\Big], \quad (9)$$

where $\varepsilon_t$ is the shock to the interested variable in a horizon *d*, and a history $\Omega_{t-1}$.

### 3.4. Econometric Methodology

When the global crisis occurred, these crises hit the Thai economy substantially and caused a major shift in the structure. Since the Thai economy is severely export-dependent, the export and output nexus also altered correspondingly. In the presence of structural breaks, the linear correlation method seems to be an insufficient model to estimate the single effect of the whole. Therefore, the TVAR model was constructed to analyze the nonlinear influence of the growth, the export, and the import. The three original datasets consisting of the real GDP, the real export, and the real import were transformed into logarithms in order to remove trends. The following notations are used hereafter: *lGDP* is the logarithm of GDP, representing the output in Thailand at time *t* (*t* = 1, ... , *T*), *lEX* is the logarithm of export, and *lIM* is the logarithm of import. Hence, the vector of endogenous variables is specified as *X* = (*lGDP, lEX, lIM*). Initially, the VEC model was constructed to carry out the investigation of the nature of the export–output nexus, as follows:

$$\Delta lGDP_t = u_1(s_t) + \sum_{j=1}^{p-1} \beth_{1,1,j}^*(s_t)\Delta lGDP_{t-j} + \sum_{j=1}^{p-1} \beth_{1,2,j}^*(s_t)\Delta lEX_{t-j} + \sum_{j=1}^{p-1} \beth_{1,3,j}^*(s_t)\Delta lIM_{t-j} - \sum_{j=1}^{r} \alpha_{1,j}(s_t)Z_{t-1,j} + \varepsilon_{1t},$$

$$\Delta lEX_t = u_2(s_t) + \sum_{j=1}^{p-1} \beth_{2,1,j}^*(s_t)\Delta lGDP_{t-j} + \sum_{j=1}^{p-1} \beth_{2,2,j}^*(s_t)\Delta lEX_{t-j} + \sum_{j=1}^{p-1} \beth_{2,3,j}^*(s_t)\Delta lIM_{t-j} - \sum_{j=1}^{r} \alpha_{2,j}(s_t)Z_{t-1,j} + \varepsilon_{2t},$$

$$\Delta lIM_t = u_3(s_t) + \sum_{j=1}^{p-1} \beth_{3,1,j}^*(s_t)\Delta lGDP_{t-j} + \sum_{j=1}^{p-1} \beth_{3,2,j}^*(s_t)\Delta lEX_{t-j} + \sum_{j=1}^{p-1} \beth_{3,3,j}^*(s_t)\Delta lIM_{t-j} - \sum_{j=1}^{r} \alpha_{3,j}(s_t)Z_{t-1,j} + \varepsilon_{3t},$$

where $u_i(s_t)$ represents the vector of interception, $\beth_1^*(s_t)$, ... , $\beth_{p-1}^*(s_t)$ defines the matrices of autoregressive parameters, $\alpha_i(s_t)$ is the adjustment coefficient in the vector error correlation model that indicates speed of adjustment upon disequilibrium correction for acquiring the long-term equilibrium steady-state position, and $\varepsilon_t$ performs a white noise vector process in $\varepsilon_t|s_t \sim (0, \sum(s_t))$. Then, the MLE is used to obtain the estimated parameters.

## 4. Data and Empirical Results

### 4.1. Dataset

The datasets employed in the analysis of the relationships between export and output comprise the quarterly observations of real GDP, real export, and real import series (seeing Figure 1) over the period from the 1990 first quarter (q1) to the 2017 fourth quarter (q4). The data regarding the real GDP series were obtained from the Office of the National Economic and Social Development Board database, while the real export and the real import series were based on the dataset enrolled by the World Development Indicators. All the variables were transformed into logarithms. The vector of endogenous variables is specified as *X* = (*lGDP*, *lEX*, *lIM*).

Table 1 summarizes the descriptive statistics covering mean–median values, maximum–minimum values, standard deviation, and normality properties. To fundamentally determine the normality, according to Huck et al. (1986), the ranges of skewness and kurtosis of these relevant data were within the range of −2.58 to +2.58, implying that the *z*-scores performed in a normal manner. However, the large Jarque–Bera test values and the small probability values mean that the null hypothesis of normality was rejected at the 5% significance level, indicating that the GDP, EX, and IM series at the current Local Currency Units (LCUs) recorded by the World Bank have a non-normal distribution. Furthermore, the unique characteristics of these three relevant datasets were determined to stabilize the variance of the time series by eliminating trend and seasonality. The *lGDP*, *lEX*, and *lIM* series fluctuate in accordance with the impact of global economic crises. Figure 2 demonstrates that the plots of the different relevant variables after being taken through the first differentiating process could not be deemed stationary due to regime shifts and structural changes. Consequenctly, nonlinear models (i.e., threshold models) are required to diagnose the export–output growth nexus.

**Table 1.** Descriptive data of the gross domestic product (GDP), export (EX), and import (IM) series at the current LCUs (million).

| Measurements | GDP | RealEX | RealIM |
|---|---|---|---|
| Mean | 1.653 | 0.033 | 0.032 |
| Median | 1.596 | 0.029 | 0.028 |
| Maximum | 2.639 | 0.062 | 0.066 |
| Minimum | 0.823 | 0.006 | 0.007 |
| Standard Deviation | 0.492 | 0.018 | 0.018 |
| Skewness | 0.251 | 0.084 | 0.317 |
| Kurtosis | 1.866 | 1.651 | 1.753 |
| Jarque–Bera Test | 7.100 | 8.551 | 9.047 |
| Probability | 0.028 | 0.013 | 0.011 |

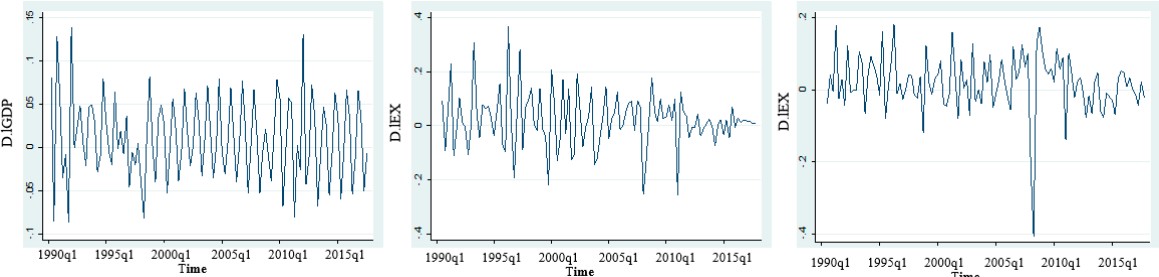

**Figure 2.** The logarithms of gross domestic product, export, and import (*lGDP*, *lEX* and *lIM*) after taking the first differentiating process.

## 4.2. Empirical Results

### 4.2.1. Unit Root and Cointegration

In order to determine whether the underlying series contained unit roots or not, the Dickey–Fuller generalized least squares (DF-GLS) test proposed by Elliott et al. (1996) was firstly conducted under the null hypothesis of a unit root. The DF-GLS method, which is transformed through a generalized least squares (GLS) for residuals, improves the power of the augmented Dickey–Fuller test in case there is a tendency to be closely stationary given that the sample size is small and an unknown mean or trend is presented. Table 2 illustrates the unit root test results. It can be clearly seen that the null hypothesis of unit roots was not rejected at all significant levels. Additionally, the DF-GLS tests conveyed that the first differences of the *lGDP, lEX,* and *lIM* were stationary, meaning that the relevant series of Thailand performed an *I(1)* process.

**Table 2.** Dickey–Fuller generalized least squares (DF-GLS) unit root test results. The "l" in front of GDP, EX, and IM represents the logarithm.

| Variables | Levels | | First Different | |
|---|---|---|---|---|
| | DF-GLS tau [a] | DF-GLS tau [b] | DF-GLS tau [a] | DF-GLS tau [b] |
| lGDP | −2.141(5) | 1.531 | −5.035 (2) *** | −5.092 (2) *** |
| lEX | −0.735 (7) | 1.032 | −3.621(7) *** | −2.760 (3) *** |
| lIM | −0.951 (8) | 0.939 | −3.095 (6) *** | −2.760 (5) *** |

*** reveals the 1% critical value. [a] A constant is involved in the test equation within the null hypothesis of nonstationarity; the critical values of 1% is equal to −3.567. [b] No trend imposed in the test equation to test stationarity around the mean instead of around the linear time trend within the null hypothesis of nonstationarity; the critical value of 1% is equal to −2.599.

For the Eigen test (cointegration test), a VAR with appropriate lag was constructed, while the lag selection criteria were Akaike information criterion (AIC), Hannan-Quinn information criterion (HQ), Schwarz Criterion (SC), and Final Prediction Error (FPE). It can be seen that the VAR model is adequate with these four lag criteria, as demonstrated in Table 3.

**Table 3.** Lag selection criteria.

| Model Selection | Lag = 1 | Lag = 2 | Lag = 3 | Lag = 4 | Lag = 5 |
|---|---|---|---|---|---|
| AIC | −15.982 | −16.494 | −16.590 | −16.681 | −16.601 |
| HO | −15.853 | −16.271 | −16.272 | −16.282 | −16.112 |
| SC | −15.670 | −15.962 | −15.834 | −15.693 | −15.394 |
| FPE | $1.151 \times 10^{-7}$ | $6.879 \times 10^{-8}$ | $6.269 \times 10^{-8}$ | $5.738 \times 10^{-8}$ | $6.223 \times 10^{-8}$ |

The Johansen eigenvalue and the trace test were employed to explore the cointegrating rank in the time-series model. Table 4 reports that the results of the eigen test and the trace test were significantly greater than the critical values. Therefore, there is strong evidence to reject the null hypothesis of no cointegration. Thus, at least one cointegrating vector appeared in the model.

**Table 4.** Johansen test output.

| Series | Hypothesis | Eigen Test | 0.05 Critical Values | Test Statistic | 0.05 Critical Values |
|---|---|---|---|---|---|
| | $r = 0$ | 14.51 ** | 9.24 | 73.59 | 34.91 |
| *lGDP, lEX, lIM* | $r \leq 1$ | 19.31 ** | 15.67 | 33.81 | 19.96 |
| | $r \leq 2$ | 39.78 ** | 22.00 | 14.51 | 9.24 |

** reveals the 5% critical value.

Thereafter, a nonlinearity test was constructed for the threshold VAR model against the linear VAR model by employing the lGDP. The threshold values imply that the turning points at which the output growth passes through are above the threshold and are connected to the expansion regime. To test the null hypothesis of linearity against the alternative of nonlinearity ($t = 1, 2$), where $t$ indicates the threshold value, a multivariate extension of the linearity test by Hansen (1999) was employed. The likelihood ratio (LR) test is formulated as follows: $LR_{01} = T\left(\ln\left(det\ \hat{\Sigma}_0\right) - \ln\left(det\ \hat{\Sigma}_1\right)\right)$, where $\hat{\Sigma}_0$, which is the estimated covariance matrix for the model, is related to the null hypothesis, and $\hat{\Sigma}_1$ is the estimated covariance matrix for the alternative hypothesis.

Table 5 summarizes the results of the threshold tests by utilizing the optimal trimming at 15%. The LR test statistics point out that the null hypothesis was rejected for all the tests. Therefore, the export–output growth for Thailand is well defined by the two-threshold VAR model.

**Table 5.** Likelihood ratio (LR) test results.

| **LR test for Linearity vs. 1 Threshold** | |
| --- | --- |
| LR statistic | 103.72 |
| *p*-Value | 0.00 |
| Estimated threshold | 10.13 (Percentage of Observations: 44.9% and 55.1%) |
| **LR test for Linearity vs. 2 Thresholds** | |
| LR statistic | 206.94 |
| *p*-Value | 0.00 |
| Estimated threshold | 10.05; 10.71 (Percentage of Observations: 40.2%, 29%, and 30.8%) |
| **LR test for 1 Threshold vs. 2 Thresholds** | |
| LR statistic | 103.23 |
| *p*-Value | 0.00 |
| Estimated threshold | 10.13; [10.05, 10.71] |

### 4.2.2. Threshold VAR (TVAR) Model

Table 6 reports that the TVAR model of the export–output nexus in Thailand is properly explained by the two-threshold VAR model or the three-regime TVAR model. The GDP ($\pi$) represents the threshold variable. In the lower regime ($\pi < 10.05$), the GDP values at the first and the fourth lag pass-through are positive and the export is negative, and these elements are statistically significant. Moreover, in the upper regime ($\pi < 10.71$), only the GDP at the fourth lag is positive and significant. However, in the middle regime ($10.5 \le \pi < 10.71$), the GDP is insignificant. These empirical findings indicate that the GDP, export, and import pass through at different stages of the business cycle. The asymmetric and manifold shocks appear differently between regimes.

Furthermore, modeling a TVAR model involves selecting the threshold variable in order to consider the number of regimes. The determination of the threshold value was carried out based on a grid search over a range of potential values of the threshold variable to govern the switching of one or more regimes. The LR statistic can be accommodated to find the appropriate multiple thresholds associated with the data. The threshold parameters were explored numerically via a grid search, and 15% of the top and the bottom were trimmed to check the processes. The estimated threshold variables were consistently found to be 10.05 and 10.71, as depicted in Figure 3.

**Table 6.** Results of threshold vector autoregressive (TVAR) model with two thresholds.

| Variables | Regime 1 $\pi(-1) \leq 10.05$ Percentage of Observations of 40.2% | Regime 2 $10.05 \leq \pi(-1) \leq 10.71$ Percentage of Observations of 29.0% | Regime 3 $\pi(-1) \geq 10.71$ Percentage of Observations of 30.8% |
|---|---|---|---|
| **lGDP** | | | |
| Intercept | −0.627 (1.945) | 3.246 (1.691) | 2.609 (1.741) |
| lGDP-1 | 0.753 (0.151) *** | 0.779 (0.464) | 0.325 (0.197) |
| lEX-1 | −0.139 (0.068) * | −0.016 (0.108) | 0.166 (0.192) |
| lIM-1 | −0.080 (0.121) | 0.056 (0.096) | −0.168 (0.202) |
| lGDP-2 | −0.102 (0.174) | −0.945 (0.476) | −0.314 (0.207) |
| lEX-2 | 0.067 (0.072) | −0.035 (0.124) | 0.119 (0.195) |
| lIM-2 | 0.139 (0.132) | 0.054 (0.115) | −0.019 (0.211) |
| lGDP-3 | 0.039 (0.172) | 0.698 (0.526) | 0.122 (0.201) |
| lEX-3 | 0.001 (0.068) | −0.008 (0.112) | −0.090 (0.184) |
| lIM-3 | −0.095 (0.129) | 0.048 (0.106) | 0.017 (0.210) |
| lGDP-4 | 0.433 (0.161) ** | 0.122 (0.441) | 0.545 (0.185) ** |
| lEX-4 | 0.147 (0.073) * | −0.018 (0.080) | 0.226 (0.209) |
| lIM-4 | −0.152 (0.108) | 0.091 (0.100) | −0.056 (0.163) |
| **lEX** | | | |
| Intercept | −5.167 (3.944) | −1.568 (3.428) | 5.558 (3.528) |
| lGDP-1 | 0.277 (0.305) | −0.281 (0.940) | 0.205 (0.399) |
| lEX-1 | 0.502 (0.138) *** | 0.638 (0.218) ** | 0.348 (0.389) |
| lIM-1 | 0.299 (0.245) | 0.433 (0.194) * | 0.658 (0.409) |
| lGDP-2 | −0.745 (0.351) * | 0.818 (0.964) | 0.566 (0.419) |
| lEX-2 | −0.214 (0.145) | 0.020 (0.251) | 0.619 (0.396) |
| lIM-2 | −0.477 (0.266) | −0.029 (0.232) | −0.431 (0.426) |
| lGDP-3 | 0.928 (0.348) ** | −0.797 (1.066) | −0.471 (0.408) |
| lEX-3 | 0.539 (0.137) *** | −0.132 (0.227) | 0.432 (0.372) |
| lIM-3 | −0.729 (0.260) ** | −0.103 (0.214) | −0.684 (0.424) |
| lGDP-4 | −0.007 (0.325) | 0.208 (0.893) | −0.116 (0.375) |
| lEX-4 | −0.048 (0.147) | 0.152 (0.163) | −0.205 (0.423) |
| lIM-4 | 1.015 (0.219) *** | 0.248 (0.204) | 0.524 (0.329) |
| **lIM** | | | |
| Intercept | 0.017 (0.022) | −0.065 (0.324) | 0.419 (0.155) |
| lGDP-1 | −8.819 (3.761) * | −2.826 (3.270) | −4.241 (3.366) |
| lEX-1 | 0.754 (0.291) * | −0.744 (0.897) | 0.056 (0.381) |
| lIM-1 | 0.210 (0.234) | 0.846 (0.185) *** | 1.315 (0.390) ** |
| lGDP-2 | −0.281 (0.335) | 1.022 (0.920) | 0.158 (0.400) |
| lEX-2 | 0.076 (0.138) | 0.0206 (0.251) | 0.661 (0.377) |
| lIM-2 | 0.012 (0.254) | 0.040 (0.221) | −0.666 (0.407) |
| lGDP-3 | 0.204 (0.332) | −0.896 (1.017) | −0.411 (0.389) |
| lEX-3 | 0.044 (0.131) | −0.004 (0.217) | −0.602 (0.404) |
| lIM-3 | −0.268 (0.248) | −0.219 (0.204) | 0.727 (0.355) * |
| lGDP-4 | 0.289 (0.310) | 0.774 (0.852) | −0.299 (0.357) |
| lEX-4 | 0.161 (0.140) | 0.040 (0.156) | 0.524 (0.329) |
| lIM-4 | 0.256 (0.209) | 0.230 (0.194) | 0.440 (0.314) |

***, **, and * reveal 1%, 5%, and 10% critical values.

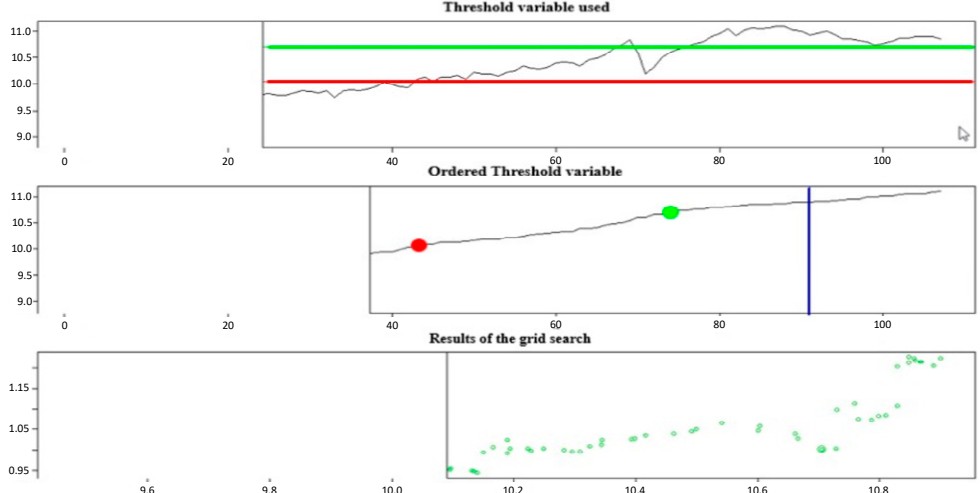

**Figure 3.** The grid search of the two-threshold vector autoregressive (VAR) model.

### 4.2.3. Threshold Vector Error Correction (TVEC) Model

The threshold vector error correction model (TVECM) with different regimes was used for the threshold estimators based on their maximum likelihood estimations. Table 7 presents the results of the two-threshold TVECM. The threshold values were −49.20 and −45.93. Both thresholds were found to be consistent with the grid search of the two-threshold TVECM, as shown in Figure 4. The percentages of observations for the lower, medium, and upper regimes were 36.8%, 33.0%, and 30.2%, respectively. The Error Correction Term (ECT) in regime 3 was negative and statistically significant, signifying the speed of the model with regard to long-term equilibrium. Moreover, all the regimes were found to respond to the deviations from the long-term equilibrium in the nonlinear model. Additionally, it was observed that the export and output nexus tends to grow at a faster speed during an expansion than during a recession.

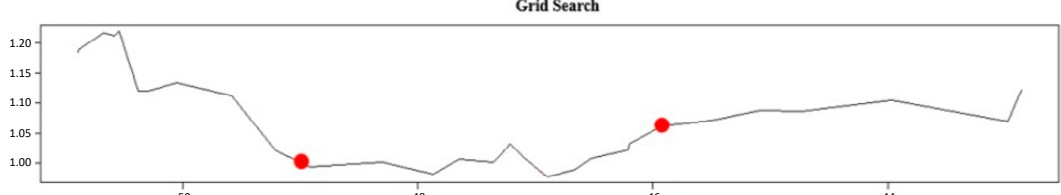

**Figure 4.** The grid search of the two-threshold vector error correction (VEC) model.

**Table 7.** Results of threshold vector error correction (TVEC) model with two thresholds.

| Variables | Regime 1<br>$\pi(-1) \leq -49.20$<br>Percentage of<br>Observations of 36.8% | Regime 2<br>$-49.20 \leq \pi(-1) \leq -45.93$<br>Percentage of<br>Observations of 33.0% | Regime 3<br>$\pi(-1) \geq -45.93$<br>Percentage of<br>Observations of 30.2% |
|---|---|---|---|
| **lGDP** | | | |
| ECT | −0.002 (0.906) | −0.004 (0.647) | −0.008 (0.021) * |
| Intercept | −0.054 (0.924) | 0.191 (0.629) | 0.391 (0.015) * |
| lGDP-1 | −0.270 (0.134) | 0.0192 (0.955) | 0.513 (0.008) ** |
| lEX-1 | 0.105 (0.462) | 0.042 (0.579) | −0.228 (0.007) ** |
| lIM-1 | −0.151 (0.279) | 0.006 (0.938) | 0.042 (0.727) |
| lGDP-2 | −0.600 (0.002) ** | −0.400 (0.320) | −0.389 (0.037) * |
| lEX-2 | 0.120 (0.500) | −0.012 (0.900) | 0.030 (0.763) |
| lIM-2 | −0.035 (0.791) | 0.040 (0.593) | 0.164 (0.267) |
| lGDP-3 | −0.300 (0.130) | −0.044 (0.904) | 0.551 (0.005) ** |
| lEX-3 | 0.016 (0.921) | 0.013 (0.853) | −0.136 (0.131) |
| lIM-3 | −0.047 (0.733) | 0.041 (0.632) | 0.095 (0.539) |
| lGDP-4 | 0.336 (0.083)8 | 0.549 (0.153) | −0.071 (0.671) |
| lEX-4 | 0.226 (0.131) | 0.040 (0.576) | 0.077 (0.380) |
| lIM-4 | 0.191 (0.500) | −0.300 (0.194) | −0.073 (0.735) |
| **lEX** | | | |
| ECT | −0.011 (0.158) | −0.039 (0.032) * | −0.038 (0.110) |
| Intercept | −0.418 (0.220) | −1.851 (0.033) * | 1.987 (0.107) |
| lGDP-1 | 0.533 (0.194 | −0.420 (0.568) | −0.326 (0.400) |
| lEX-1 | −0.092 (0.608) | −0.194 (0.237) | −0.635 (0.042) * |
| lIM-1 | 0.210 (0.421) | 0.534 (0.002) ** | 0.364 (0.228) |
| lGDP-2 | −0.261 (0.512) | −0.331 (0.702) | 0.422 (0.280) |
| lEX-2 | −0.288 (0.186) | −0.442 (0.010) * | −0.344 (0.328) |
| lIM-2 | −0.449 (0.162) | 0.039 (0.807 | 0.358 (0.214) |
| lGDP-3 | 0.987 (0.019) * | −0.502 (0.527) | −0.224 (0.599) |
| lEX-3 | −0.226 (0.152) | −0.227 (0.152) | 0.016 (0.929) |
| lIM-3 | 0.095 (0.538) | 0.041 (0.632) | −0.283 (0.343) |
| lGDP-4 | −0.071 (0.671) | −0.786 (0.345) | −0.201 (0.629) |
| lEX-4 | 0.203 (0.286) | −0.172 (0.266) | −0.377 (0.243) |
| lIM-4 | 0.309 (0.291) | 0.498 (0.038) * | 0.248 (0.268) |
| **lIM** | | | |
| ECT | −0.161 (0.542) | −0.531 (0.202) | −0.322 (0.282) |
| Intercept | −0.188 (0.136) | −0.242 (0.310) * | −0.144 (01328) |
| lGDP-1 | 0.335 (0.399) | 1.062 (0.146) | −0.054 (0.885) |
| lEX-1 | 0.110 (0.599) | 0.401 (0.014) * | −0.132 (0.659) |
| lIM-1 | −0.445 (0.081) | −0.301 (0.063) | −0.007 (0.980) |
| lGDP-2 | −0.002 (0.994) | 1.895 (0.027) * | 0.227 (0.547) |
| lEX-2 | 0.110 (0.599) | −0.442 (0.010) * | 0.024 (0.941) |
| lIM-2 | −0.349 (0.260) | −0.574 (0.000) *** | 0.202 (0.468) |
| lGDP-3 | 0.049 (0.901) | 1.232 (0.113) | 0.096 (0.815) |
| lEX-3 | 0.030 (0.869) | 0.209 (0.172) | 0.520 (0.129) |
| lIM-3 | −0.448 (0.167) | −0.657 (0.000) *** | −0.353 (0.222) |
| lGDP-4 | 0.245 (0.482) | 2.380 (0.004) ** | −0.258 (0.521) |
| lEX-4 | −0.015 (0.932) | 0.354 (0.019) * | 0.103 (0.741) |
| lIM-4 | 0.191 (0.500) | −0.300 (0.193) | −0.073 (0.735) |

***, **, and * reveal 1%, 5%, and 10% critical values.

### 4.2.4. Linear Cointegration and Threshold Cointegration Models

In this approach, it was examined whether the relationship of the export–output growth nexus can be appropriately characterized by the linear cointegration or threshold cointegration. To follow Krolzig (1997), a test was carried out within the null hypothesis of linear cointegration against threshold

cointegration by implementing fixed regressors and residual bootstrap. Initially, the cointegrating value was estimated using the linear VECM. Then, the sup-LM test was operated for an array of different threshold values. The parametric residual bootstrap executed 100 simulation replications. The maximum values of those sup-LM tests are reported in Table 8.

**Table 8.** Results of linear vs. threshold cointegration.

| Test | lGDP & lEX | | lGDP & lIM | | lEX & lIM | |
|---|---|---|---|---|---|---|
| **Sup-LM test** (*p*-Value) | 18.763 * (0.002) | Critical value 16.311 (90%) 17.545 (95%) 19.838 (99%) | 18.927 * (0.004) | Critical value 16.481 (90%) 18.164 (95%) 23.209 (99%) | 14.826 * (0.21) | Critical value 16.212 (90%) 17.470 (95%) 17.870 (99%) |
| **Maximized threshold value** | 9.472 | | 9.296 | | −0.964 | |
| **Cointegrating value** | −0.467 | | −0.487 | | −1.107 | |
| **Number of bootstrap replications: 100** (fixed-repressor bootstrap) | | | | | | |

* Significant at the $p < 0.01$ level.

The sup-LM test statistics indicate that the null hypothesis of linearity was rejected at the 90% and the 95% confidence levels for lGDP & lEX and lGDP & lIM, in favor of the threshold cointegration models, while the null hypothesis for the lEX & lIM test was not rejected. Consequently, this test verifies that the cointegration between the GDP and the export, as well as the cointegration between the GDP and the import, appears to be in large part due to adjustment in the export–output growth nexus. On the other hand, the export and the import do not appear to respond to the adjustment. Figure 5 depicts the results of the tests of linear vs. threshold cointegration and the density of bootstrap, which proved to be normal distributions for all the relevant variables.

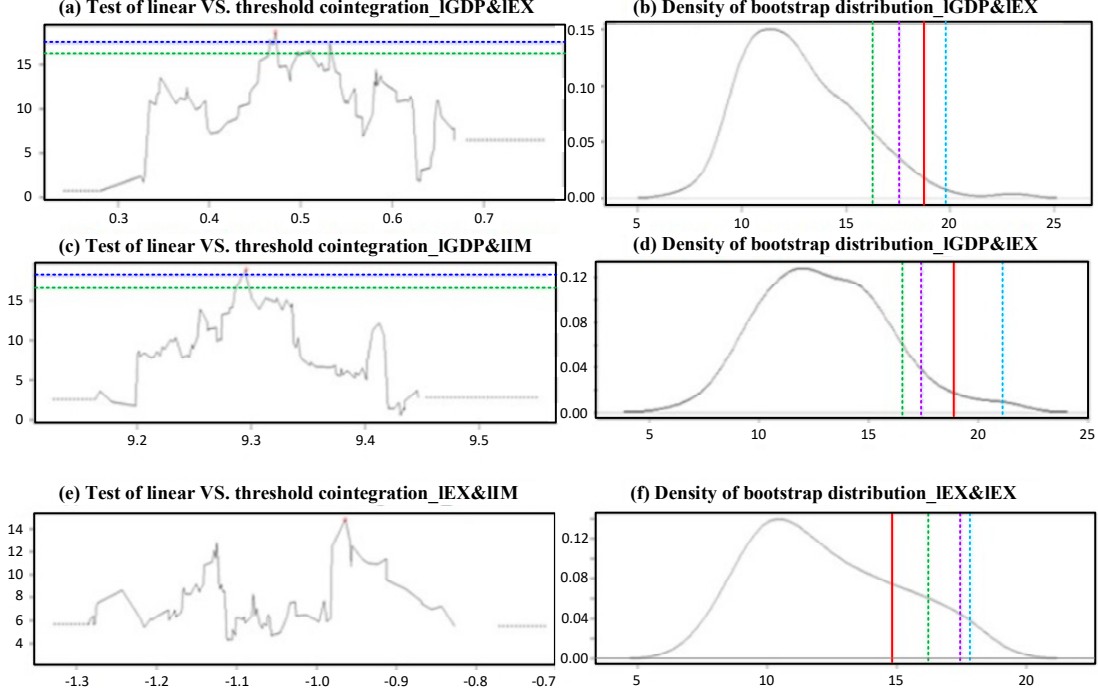

**Figure 5.** Results of the tests of linear vs. threshold cointegration.

In the context of information criterion statistics, the Akaike information criterion (AIC) and the Bayesian information criterion (BIC) were adapted as measures for model selection. The TVECM within the two thresholds provided evidence in favor of the ELG hypothesis in the three regimes on account of the lower AIC and BIC values, as shown in Table 9. Consistently, the test of threshold cointegration was preferred. Moreover, this study demonstrated that export and import are the key driving forces behind economic growth as far as Thailand is concerned.

**Table 9.** Results of linear vs. threshold cointegration. AIC—Akaike information criterion; BIC—Bayesian information criterion.

| Model Selection | Linearity | Nonlinearity | | | |
|---|---|---|---|---|---|
| | VECM | TVAR (1 Threshold) | TVAR (2 Thresholds) | TVEC (1 Threshold) | TVEC (2 Thresholds) |
| AIC | −1566.586 | −1792.417 | −1801.531 | −1786.440 | −1790.023 |
| BIC | −1496.851 | −1628.807 | −1483.465 | −1560.048 | −1449.103 |

### 4.2.5. Five-Year Forecasting of TVAR

Next, the level of a series estimated through a two-threshold VAR within a four-lag length was predicted. The in-sample data (1990–2012) were utilized to run the estimation, while the out-sample data (2013–2017) were used to produce the forecast. The unique threshold values were 14.283 and 14.504 (AIC = 0.587, 0.581). A comparison was made between the traditional VEC and the two-threshold VAR models. The root-mean-square error (RMSE) and the mean percentage error (MPE) were used to evaluate the predictive power across those forecasting models. As for assessing the performance of the prediction models, the errors for the two-threshold VAR model were much smaller (RMSE = 0.044 and MAE = 0.033) than for the traditional VEC model (RMSE = 0.064 and MAE = 0.042). As a consequence, the two-threshold VEC within the four-lag length had better prediction accuracy of the model due to the lower values of RMSE and MPE.

Thereafter, a five-year forecast was produced. Figure 6 displays the five-year forecast for the export–output nexus in Thailand over the period 2018–2022. The forecasts with regard to the economy of the export–output nexus upswing show that it will continue at a slower pace for the GDP, which will enable the maintenance of its strong growth pace during the next three years. Nevertheless, the export and import scenario will fluctuate at a higher pace from the period of 2019–2022 onward.

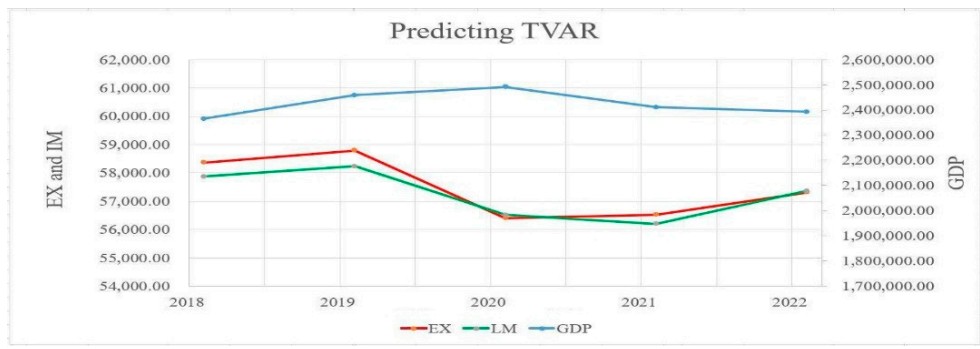

**Figure 6.** The five-year forecasting of TVAR.

### 4.2.6. Generalized Impulse Response Function (GIRF)

The GIRF can measure influences of the shock towars the behavior of a system within a given horizon. With the nonlinear generalized IRF specification, it can capture state asymmetry by inquiring the distinct effect of a shock depending on the regime in which it appears, while also taking into account the unconditional response of a series to this break. As such, we produce the regime-IRF to

determine the persistence at stake within the regime due to future structure breaks. Figure 7 presents the GIRF when shocks, which correspond to modifications in *lGDP*, *lEX*, and *lIM*, occur in regime 1 (up), regime 2 (medium), and regime 3 (down).

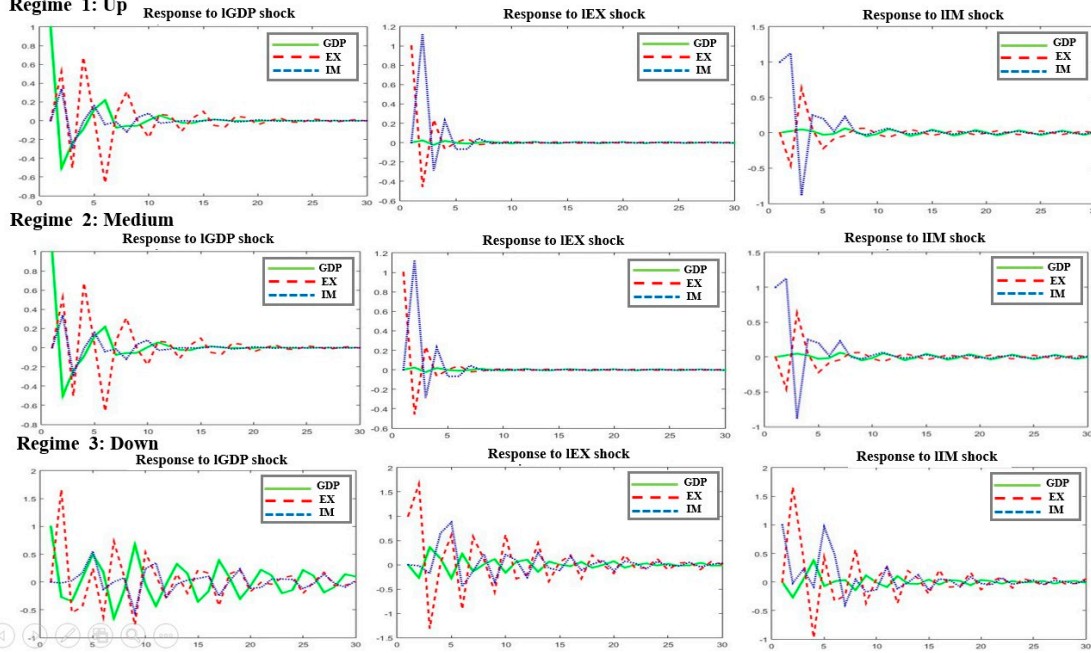

**Figure 7.** Generalized impulse response function (GIRF) due to shocks in regimes 1, 2, and 3.

We can see that, when the shocks appear in the prevailing regime, the long-term persistence is distinctly different across these three regimes. The GIRF is less persistent under regime 3 than the high regimes 1 and 2. For regimes 1 and 2, changes in the GDP are contemporaneously reflected in a reduction of economic growth, while the GDP has a positive effect on the export and output nexus. Furthermore, the export and import nexus reacts to movements in the short term for all variables of the system. Consistently, for regime 3, once the shocks appear, these three variables fluctuate more in the first half of the period and ultimately reach convergence levels. The important reason why the nexus between exports and imports affects the economic growth is that Thailand currently enhances and promotes intensive trade liberalization. The exports and imports are now an essential part of economic growth. As the GDP expands, the demand for goods increases, which in turn leads to a raise in the exports and imports.

## 5. Conclusions and Policy Recommendations

This paper examined the ELG hypothesis accounting for the business cycle asymmetry of Thailand. The threshold VAR and VEC models were applied to investigate the relationship between the GDP, the export, and the import during the period from 1990 to 2017 in Thailand. The crucial conclusions drawn can be summarized as follows:

- The export–output behavior responding to economic crises can be reasonably characterized by the two-threshold VAR model (the threshold values were 10.05 and 10.71) and the two-threshold VEC model (the threshold values were −49.20 and −45.93). These export–output scenarios are distorted because of the long-term equilibrium. The empirical findings convey that the export, import, and output pass-throughs at the lower, medium, and upper regimes react to changes in the structure. Those structural breaks have a significant effect on the different regimes. Consequently, for Thailand, this phenomenon supports the ELG hypothesis with regard to business cycle asymmetry.

- According to threshold cointegration, the GDP associated with the export adjusts the long-term equilibrium relationship; however, export and import do not exist with regard to the long-term equilibrium relationship when business cycles take place.
- As for the evaluation of the performance of the prediction models, the two-threshold VAR model has a better goodness-of-fit measurement because of the smaller RMSE and MPE values than the VEC model. Thereafter, the five-year forecast (2018–2022) was predicted. The export–output growth nexus scenarios seem to undergo a consistent upswing.
- The regime IRFs allowed us to explicitly identify the reactions of the whole system at stake within each regime (up, medium, and down) in response to structure breaks. These dynamics are quite different across regimes.

This paper has some policy implications. Thailand's economy can progressively encourage a shift toward outward-oriented trade policies. As a consequence, Thailand is severely reliant on export-led growth strategies. Simultaneously, the ELG strategies can be implemented to seek importing of capital goods, intermediate goods, and raw materials or commodities at cheaper prices somewhere else, as made evident by the perspicuity of the ELG hypothesis. Hence, the import sectors also play a crucial role for the export–output growth nexus in Thailand's economy. The government should highlight the balance of international trade to accomplish a trade surplus in order to keep the strength of the country's economy. Consequently, the government should create political and economic stability, with the adoption of export-promoting trade strategies and import-substituting policies. These effective strategies bring more credibility of a country toward international competition. Ultimately, a country can reach balanced growth for the long term.

However, because boom and bust economic cycle occurrences are a reality, policy-makers should emphasize the role of countercyclical macroeconomic policies. The appropriate countercyclical policies can encourage the strengthening of recovery regimes and stimulation of shorter recession regimes. Based on the results shown here, the export–output growth nexus should be categorized at the lower, medium, and upper states.

Furthermore, as seen in the five-year economic forecasts of the GDP, export, and import growth, these factors seem to affect economy-wide fluctuations around a short-term growth trend. Recessions have great influence and can cause tremendous disadvantages such as crises and sharp economic downturns. Hence, stabilization and consistent outward-oriented trade policies should be implemented, as they are associated with the speed of economic recovery and its economic potential with respect to the Thailand economy.

**Author Contributions:** A.R. was the main writer of the paper, collected data, and analyzed the results. J.L. constructed the model analysis and developed the source coding for the R software. S.S. supervised the overall research and revised the empirical findings and policy recommendations. All authors revised and approved the final manuscript.

**Funding:** This research received no external funding.

**Conflicts of Interest:** The authors declare no conflict of interest.

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
