# Peer review of "Export–Output Growth Nexus Using Threshold VAR and VEC Models: Empirical Evidence from Thailand"

_economies, doi:10.3390/economies7020060_

Author Response

 a point-by-point response to the reviewer’s comments has already done as the attached file

Reviewer 2 Report

Dear Author(s),

Your work is written with good intent and used good methods. However, the presentation was not standard for a journal article. My major issues are: First, you did not tell us how your contribution is important. Why should we care about it? You need to explain for a reader to be convinced the work is important. Second, you presented your econometric methods in the section Background rather than in the econometric methodology section. In the background section, you need to present the literature about the background of the issue and not econometric methods. Please rewrite this. Bring in the background meat and not econometrics in this section. Third, the framework that you put in Econometric Methodology section is actually Empirical Specification section and not how you called it. Please rewrite this. In addition, when you presented your methods, you did not tell us why used opted for these methods, you just started describing them. You lose the reader like this. Furthermore, your data section is so short. Try to present the descriptive analyses of your data so we have an overview of what to expect in results. Is this the reason you combined data with empirical results? This is nor correct. Please separate the two issues! Minor comments are in your document attached. I added them in your main document as notes.

Thank you!

Author Response

 a point-by-point response to the reviewer’s comments has already done as the attached file

Round  2

Reviewer 2 Report

You made relevant changes as suggested. However, the English is poorly written. Even your responses to my requests were written with poor grammar. See the attached manuscript. I have painted where your English is not sound. Please correct! For example, line 44 on page 2, a hypothesis doesn't testify, it is tested! Line 56 on page 2, you don't say: Each phases, and so many more instances. Please correct!

Author Response

a point-by-point response to the reviewer 2’s comments is adjusted as the attached file.
